# Hard Example Generation by Texture Synthesis for Cross-domain Shape Similarity Learning

**Huan Fu**[1*]    **Shunming Li**[1*]    **Rongfei Jia**[1]    **Mingming Gong**[2]

**Binqiang Zhao**[1]    **Dacheng Tao**[3]

[1]TaoXi Technology Department, Alibaba Group, China
[2]The University of Melbourne, Australia
[3]The University of Sydney, Australia

{fuhuan.fh, shunming.lsm, rongfei.jrf, binqiang.zhao}@alibaba-inc.com
mingming.gong@unimelb.edu.au  dacheng.tao@sydney.edu.au

## Abstract

Image-based 3D shape retrieval (IBSR) aims to find the corresponding 3D shape of a given 2D image from a large 3D shape database. The common routine is to map 2D images and 3D shapes into an embedding space and define (or learn) a shape similarity measure. While metric learning with some adaptation techniques seems to be a natural solution to shape similarity learning, the performance is often unsatisfactory for fine-grained shape retrieval. In the paper, we identify the source of the poor performance and propose a practical solution to this problem. We find that the shape difference between a negative pair is entangled with the texture gap, making metric learning ineffective in pushing away negative pairs. To tackle this issue, we develop a geometry-focused multi-view metric learning framework empowered by texture synthesis. The synthesis of textures for 3D shape models creates hard triplets, which suppress the adverse effects of rich texture in 2D images, thereby push the network to focus more on discovering geometric characteristics. Our approach shows state-of-the-art performance on a recently released large-scale 3D-FUTURE [1] repository, as well as three widely studied benchmarks, including Pix3D [2], Stanford Cars [3], and Comp Cars [4]. Codes will be made publicly available at: https://github.com/3D-FRONT-FUTURE/IBSR-texture.

## 1 Introduction

Cross-domain image-based 3D shape retrieval (IBSR) is to identify the matched 3D CAD model of the object in a query image. With the growing number of 3D shapes, the establishment of reliable IBSR platforms is significant. For example, it can help to build 3D virtual scenes for real-world houses from 2D images. High-performing IBSR systems may also inspire and benefit studies in shape collection based 3D object reconstruction. Unluckily, compared to content-based image retrieval [5, 6, 7, 8, 9], IBSR is challenging due to the large appearance gap between 3D shapes and 2D images.

There are mainly two directions to alleviate the issue, as shown in Fig. 1. For the first one, researchers take efforts to predict 2.5D sketches from images, such as surface normal, depth, and location filed, to bridge the gaps between 3D and 2D domains [2, 10, 11, 12, 13, 14, 15]. These methods achieve

---

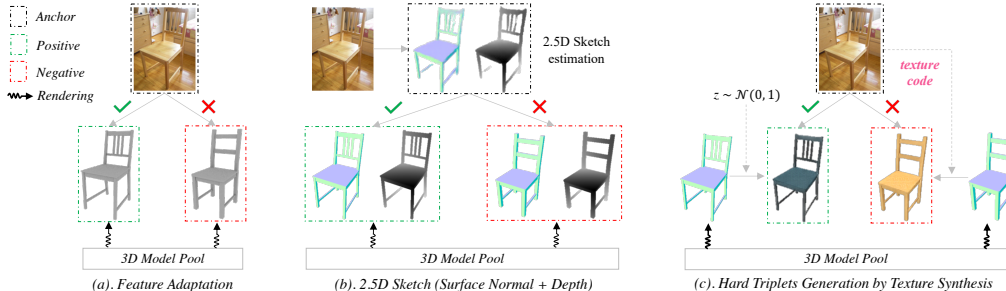

Figure 1: **Motivation.** We show the differences (based on metric learning) between our solution (c) and two widely studied routines ((a) and (b)) for cross-domain image-based 3D shape retrieval. $z$ is a random latent code. Our method assigns the "yellow" texture to the negative 3D chair, and synthesis a random texture ("blue") to the positive 3D chair. Since the anchor image also contain a "yellow" texture, the learning procedure may more easily suppress the adverse effects of creamy texture in 2D images, thereby push the network to focus more on discovering geometric characteristics.

state-of-the-art (SOTA) retrieval accuracy and sequences. However, while 2.5D representations for 3D CAD models can be conveniently rendered, accurately estimating 2.5D sketches from 2D images is itself an open problem [16, 17, 18, 19, 20, 21]. The second routine is to map cross-domain representations into a unified constrained embedding space and define (or learn) a shape similarity measure [22, 23, 24, 25, 26, 27, 28, 29, 30, 31, 32, 33]. Standard metric learning with feature adaptation techniques seems to be a natural solution to shape similarity learning. However, their performance is often unsatisfactory for fine-grained shape retrieval. One of the possible reasons is that the shape difference between a negative pair is entangled with the texture gap, making metric learning ineffective in pushing away negative pairs.

As the analysis, assuming that we have many hard triplets like $(x, x^+, x^-)$, where the positive example $x^+$ and the anchor image $x$ are identical in shape, but differ in texture, the negative example $x^-$ and $x$ are similar in texture, but differ in shape. The metric learning procedure thus should suppress the adverse impact of texture in 2D images, thereby push the network to focus more on discovering geometric characteristics. Towards the goal, we develop a geometry-focused multi-view metric learning framework empowered by texture synthesis. The synthesis of textures for 3D models creates hard triplets. More specifically, we learn a conditional generative adversarial network that can assign a texture to a 3D model based on a texture code encoded from an example image. Then, we can generate hard samples in an online manner to improve the cross-domain shape similarity learning, as shown in Fig 1. Note that, in contrast to category-driven metric learning for image retrieval, each 3D shape is an individual instance (category). It's not easy to apply hard sample mining strategies in image retrieval to IBSR. We are the first to consider texture information to generate hard triplets based on the special properties of the IBSR subject.

Furthermore, the geometry-focused multi-view metric learning framework also contains: (1) a mask attention mechanism to enhance the feature from saliency regions, and (2) a simple but effective viewpoint attention policy to guide the multi-view embedding aggregation. With these guidances, we find that the network can easily focus more on the learning of shape characteristics. Our approach shows state-of-the-art performance on a recently released large-scale 3D-FUTURE [1] repository, as well as three widely studied benchmarks, including Pix3D [2], Stanford Cars [3], and Comp Cars [4]. We also conduct various ablation studies to comprehensively discuss our method.

## 2 Proposed Method

Our main goal is to learn an encoder that can map both natural query images and 3D shapes to an embedding space. We compute the distances between a query image and the 3D shapes to perform cross-domain retrieval. To match the 3D shapes with the 2D images, we render multiple views (12 views in our paper) of surface normal $S$ for each 3D model using Blender [34]. During the learning procedure, we have $N$ query image and multi-view surface normal pairs $\{(X^i, S^i)\}_{i=1}^N$. Each query image contains a ground-truth mask $M$, a viewpoint label $V$, and a category label $C$ (*e.g.,* sofa, chair, and table). Each 3D shape is coupled with a UV atlas (texture map), so that we can also render

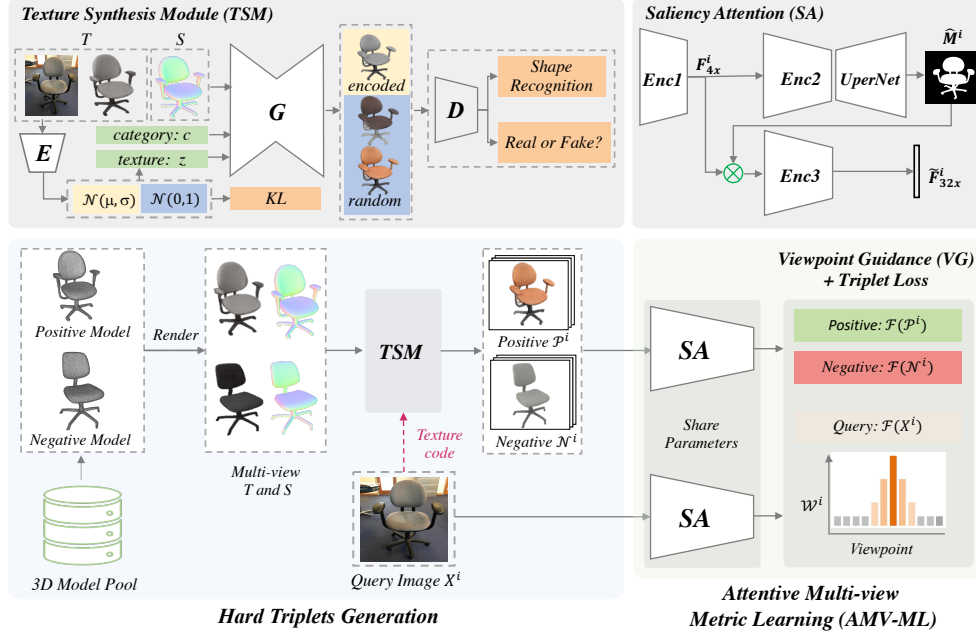

Figure 2: Bottom: The main pipeline of our method. Top: The architectures of some components in our framework. We generate hard triplets in an online manner to empower our shape similarity learning network. See Sec. 2 and Fig. 1 for more details.

synthetic 2D image $T$ (with the corresponding texture) from a random viewpoint. Note that, though public benchmarks [35] provide Shape-UV pairs, there are some reasons that we can not directly recast IBSR as an standard image retrieval problem. Firstly, large amounts of shapes in practice do not have the corresponding UV atlases. For example, while ShapeNetCore [35] contains 51,300 3D models, only several thousands shapes contain UV atlases. Secondly, we concern shape similarities rather than texture similarities in IBSR. However, texture information is a core feature for typical image retrieval subjects. These special properties motivate us to generate hard triplets as shown in Figure 1. Our full framework mainly contains a texture synthesis module (TSM) for hard examples generation and an attentive multi-view metric learning (AMV-ML) process, as shown in Figure 2. In the following, we will first introduce how do we generate hard examples by texture synthesis in an online manner. Then, we will present the attention mechanisms in the metric learning framework.

## 2.1 Hard Example Generation

Our hard example generation policy relies on texture synthesis. If ignoring the texture consistency on 3D surfaces, texture synthesis can be recast as a special image-to-image (I2I) translation task [36, 37, 38, 39, 40, 41, 42, 43, 44, 45, 46, 47, 48, 49]. For instance, VON [50] demonstrated that integrating 2D texture priors into CycleGAN [36] could make controllable texture generation possible. [51, 52, 53, 54, 55] studied conditional GANs for sketch-to-image synthesis according to example texture and color.

Specific to the particular properties of the IBSR subject, we are the first to consider texture information for hard triplets generation to improve the shape similarity learning. Thus, we do not focus on investigating the high-quality UV mapping on a 2D texture atlas and a parameterized 3D mesh. This is out of the scope of the paper. Instead, we intuitively study the objectives of Variational Autoencoder (VAE) and Generative Adversarial Network (GAN) in a hybrid model, and extend it to a conditional scenario (cVAE-GAN) [50, 53] for our purpose. Further, we improve cVAE-GAN to remedy the category-specific training issue and enhance the controlment of the example texture.

**Beyond cVAE-GAN.** The baseline cVAE is to firstly capture a latent code $z$ from a textured image $T$ by learning a encoder $E$, then adopt a generator $G$ to obtain $\widetilde{T}$ (a reconstruction of $T$) from $(S, z)$:

$$z \sim E(T) = Q(z|T), \quad \widetilde{T} \sim G(S, z) = P(T|S, z). \tag{1}$$

Here, $S$ is the rendered surface normal of a specific 3D shape from a random camera pose (6DoF), $T$ is the corresponding 2D image (with texture) under the same pose, and $z \sim E(T)$ is sampled using a re-parameterization trick. GAN [56] performs an minimax game to ensure $\widetilde{T}$ and $T$ distinguishable in both style and content. As an extension, we first render a $T'$ that differs from $T$ in the camera pose, and additional enforce $G$ to reconstruct $T$ from $(S, z')$. We find that this will intensify the texture guidance of the reference (or example) texture image.

Another issue in cVAE-GAN for texture synthesis is that it may generate textures with unreasonable part segmentation for the objects when we have mixture categories of 3D shapes in our training set. Some cases are shown in Figure 3. It seems like previous works ignore the problem by training category-specific texture synthesis models [50, 53]. We provide a partially solution by injecting a semantic latent code $c$ when learning $G$, such that $\widetilde{T} \sim G(S, z, c) = P(T|S, z, c)$. We estimate $c$ from $S$ via a pre-trained ResNet-18 for shape classification. The network is fixed during the training of our TSM. Furthermore, we incorporate an auxiliary supervision (shape recognition) on top of the discriminator $D$ to impose the category explainable texture synthesis.

**Objectives.** As the analysis, the adversarial loss of our TSM takes the form as:

$$\begin{aligned} \mathcal{L}_{adv} = \mathbb{E}_T[\log(D(T))] + 0.5 * \mathbb{E}_{(S, z \sim E(T), c)}[\log(1 - D(G(S, z, c)))] \\ + 0.5 * \mathbb{E}_{(S, z' \sim E(T'), c)}[\log(1 - D(G(S, z', c)))]. \end{aligned} \quad (2)$$

We additionally apply the $l_1$ loss to enforce the reconstruction process:

$$\mathcal{L}_{rec} = \mathbb{E}_{(T, S, z \sim E(T), c)}[\|G(S, z, c) - T\|_1] + \mathbb{E}_{(T, S, z' \sim E(T'), c)}[\|G(S, z', c) - T\|_1]. \quad (3)$$

To allow texture synthesis from a random latent code at the inference stage, we utilize a Kullback–Leibler (KL) loss to push $E(T)$ to be close to a Gaussian distribution:

$$\mathcal{L}_{KL} = \mathbb{E}_T[\mathcal{D}_{KL}(E(T)\|\mathcal{N}(0, 1))] + \mathbb{E}_{T'}[\mathcal{D}_{KL}(E(T')\|\mathcal{N}(0, 1))], \quad (4)$$

where $\mathcal{D}_{KL}(p\|q) = -\int p(z) \log \frac{p(z)}{q(z)} dz$. We further adopt a cross-entropy loss to mitigate the unexpected cross-category synthesis issue:

$$\begin{aligned} \mathcal{L}_{cls} = \mathbb{E}_T[-\log(D_{cls}(C|T))] + 0.5 * \mathbb{E}_{(S, z, c)}[-\log(D_{cls}(C|G(S, z, c)))] \\ + 0.5 * \mathbb{E}_{(S, z', c)}[-\log(D_{cls}(C|G(S, z', c)))]), \end{aligned} \quad (5)$$

where $D_{cls}$ shares the same parameters with the discriminator $D$ in the convolutional part, and $C$ is the category label of the 3D shape.

The full objective for the texture synthesis module is written as:

$$\mathcal{L}_{TSM} = \mathcal{L}_{adv} + \lambda_{rec}\mathcal{L}_{rec} + \lambda_{KL}\mathcal{L}_{KL} + \lambda_{cls}\mathcal{L}_{cls}. \quad (6)$$

The hyperparameters $\lambda_{rec}$, $\lambda_{KL}$, and $\lambda_{cls}$ are set to 10.0, 0.01, and 1.0 in all the experiments.

## 2.2 Attentive Multi-View Metric Learning (AMV-ML)

For conventional presentation, assuming that we have $N$ hard triplets $\{(X^i, \mathcal{P}^i, \mathcal{N}^i)\}_{i=1}^N$, where $\mathcal{P}^i = G(S^i, z, c^i)$, and $\mathcal{N}^i = G(S^j, E(X^i), c^j)$. Here, $S^i$ pairs up with $X^i$ in shape, and $S^j$ $(j \neq i)$ is a random negative sample from $\{S_k\}_{k=1}^N$. See Sec. 2.1 for details of the texture generator $G$ and other variables. Directly learning a multi-view metric learning based embedding network would be a straightforward solution to IBSR. We go a further step by introducing attention mechanisms to address the background noises and viewpoint variation issues of query images.

**Saliency Attention (SA).** We introduce a saliency attention (SA) policy to suppress the surrounding noises of objects in natural images. Specifically, we take the ResNet backbone as an encoder, and split it into two parts $Enc1$ and $Enc2$. Here, $Enc1$ is the beginning convolutional blocks until the 4x output layer (*e.g.*, *layer*2 of ResNet serials), and $Enc2$ represents the reminder convolutional blocks. Given a query image $X^i$, we can obtain several side feature maps $\{F_{4x}^i, F_{8x}^i, F_{16x}^i, F_{32x}^i\} = Enc2(Enc1(X^i))$, where $F_{4x}^i$ is the output of $Enc1(X^i)$. We add a light version of UperNet [57, 58] decoder ($Dec$) on top of $Enc1$ to predict saliency regions $\hat{M}^i = Dec(F_{4x}^i, F_{8x}^i, F_{16x}^i, F_{32x}^i)$ in a supervised manner. The objective is expressed as:

$$\mathcal{L}_{SA} = \frac{-1}{N * H * W} \sum_{i=1}^N \sum_{h,w} m_{hw}^i \log(\hat{m}_{hw}^i) + (1 - m_{hw}^i) \log(1 - \hat{m}_{hw}^i), \quad (7)$$

where $H * W$ is the spatial size of $\hat{M}^i$, $\hat{m}^i_{hw} \in \hat{M}^i$ is the probabilistic estimation in position $(h, w)$, and $m^i_{hw} \in M^i$ is the ground truth. All the images are resized to $224 \times 244$ in our experiments.

Then, we operate element-wise multiplication for each channels between $F^i_{4x}$ and $\hat{M}^i$ to capture the attention feature, and feed it to another encoder $Enc3$ to secure the enhanced feature $F^i_{32x}$:

$$\widetilde{F}^i_{32x} = Enc3(F^i_{4x} \otimes \hat{M}^i), \tag{8}$$

where $Enc3$ has the same network structure with $Enc2$. The saliency attention mechanism is simple but effective to enforce the network taking care of region of interest (RoI) for natural query images.

**Viewpoint Guidance (VG).** We incorporate soft viewpoint guidance into the multi-view metric learning procedure to mitigate the impact of unconstrained viewpoints. Specifically, we simply add an azimuth (viewpoint) classifier (driven by a cross-entropy loss $\mathcal{L}_{view}$) on top of $Enc2$ to capture the probability attention vector $\mathcal{W}^i \in R^V$ from $F^i_{32x}$. Here, $V$ denotes the predefined bin number of azimuth. Simultaneously, we use an additional convolutional layer followed by an average pooling operation to obtain the embeddings $(\mathcal{F}(X^i) \in R^d, \mathcal{F}(\mathcal{P}^i) \in R^{V \times d}, \mathcal{F}(\mathcal{N}^i) \in R^{V \times d})$ from their corresponding SA features, where $d$ is the feature dimension ($V = 12, d = 256$ in our paper).

Then, we operate $\mathcal{W}^i$ on $\mathcal{F}(\mathcal{P}^i)$ to capture the guided embedding $\widetilde{\mathcal{F}}(\mathcal{P}^i)$ of the positive sample:

$$\widetilde{\mathcal{F}}(\mathcal{P}^i) = \mathcal{F}(\mathcal{P}^i) \otimes \mathcal{W}^i = \sum_{v=1}^{V} \mathcal{F}(p^i_v) * w^i_v, \tag{9}$$

where $w^i_v$ is the $v$th attention score of $\mathcal{W}^i$. $\mathcal{P}^i = G(S^i, z, c^i)$ contains $V$ view of image $\{q^i_v\}_{v=1}^V$, as $S^i$ is the multi-view surface normal maps of shape $i$. $\widetilde{\mathcal{F}}(\mathcal{N}^i)$ can be produced in the same manner.

We now have a query image embedding $\mathcal{F}(X^i)$, the positive 3D model embedding $\widetilde{\mathcal{F}}(\mathcal{P}^i)$, and the negative 3D model embedding $\widetilde{\mathcal{F}}(\mathcal{N}^i)$. The objective for the viewpoint guided metric learning is:

$$\mathcal{L}_{VG} = \frac{1}{N} \sum_{i=1}^{N} max(\| f(\mathcal{F}(\mathcal{X}^i)) - f(\widetilde{\mathcal{F}}(\mathcal{P}^i)) \|^2 - \| f(\mathcal{F}(\mathcal{X}^i)) - f(\widetilde{\mathcal{F}}(\mathcal{N}^i)) \|^2 + \alpha, \ 0), \tag{10}$$

where $f(\cdot)$ is a $l2$ norm function as discussed in [5], and $\alpha = 0.1$ is the margin.

**Full Objective.** Apart from the loss terms mentioned above, we utilize an instance classification loss $\mathcal{L}_{inst}$ to capture shape similarity among 3D CAD instances, as previously [5, 15]. The full objective of our attentive multi-view metric learning takes the form as:

$$\mathcal{L}_{AMV-ML} = \mathcal{L}_{inst} + \lambda_1 \mathcal{L}_{view} + \lambda_2 \mathcal{L}_{SA} + \lambda_3 \mathcal{L}_{VG}, \tag{11}$$

where $\lambda_1 \sim \lambda_3$ are the trade-off hyperparameters. According to the difficulty level of optimizing each task, we set $\lambda_1 = 0.5$, $\lambda_1 = 1.0$, $\lambda_3 = 3.0$ in all of our experiments.

## 3 Experiments

In the section, we conduct extensive experiments to demonstrate the superiority of our method. We first present the implementation details, then make comparisons against previous works on four challenging benchmarks including 3D-FUTURE [1], Pix3D [2], Stanford Cars [3], and Comp Cars [4]. We do not evaluate our method on PASCAL3D+ [59] and ObjectNet3D [60] since the 3D models are not matched the objects in the images as stated in [2]. Finally, we examine various ablations to comprehensively discuss our method.

### 3.1 Implementation Details

We refer the supplemental materials for more implementation details. Codes will be shared.

**Network Architecture.** For the texture synthesis module (TSM), the encoder $E$ consists of five residual blocks followed by a liner layer. The generator $G$ follows the same architecture as U-Net [61]. We take PatchGANs [37] as our discriminator $D$ to identify in two scales ($70 \times 70$ and $140 \times 140$). The shape classifier $D_{cls}$ shares the same parameters with $D$ in the convolutional part. We add the

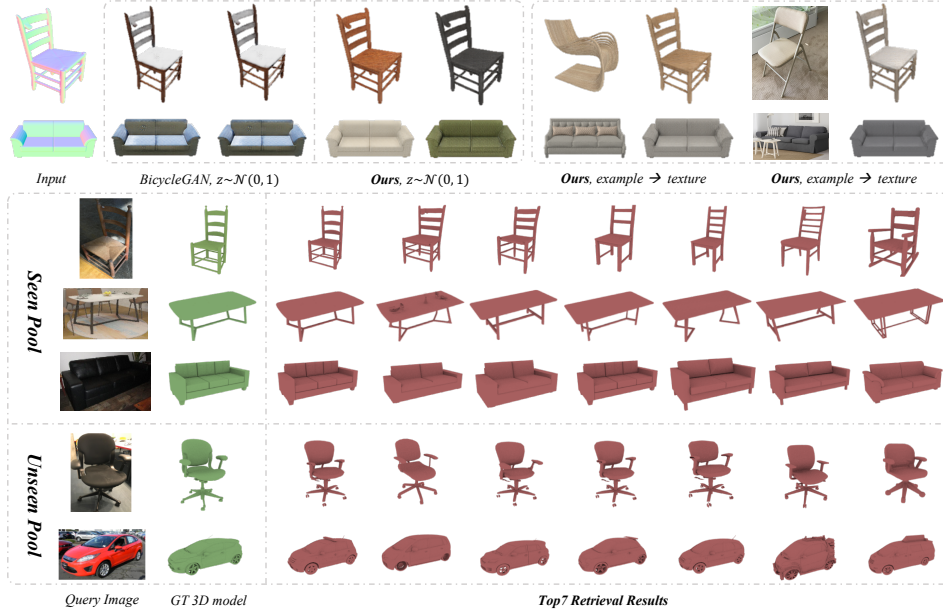

*Input*     *BicycleGAN, $z \sim \mathcal{N}(0,1)$*     **Ours**, *$z \sim \mathcal{N}(0,1)$*     **Ours**, *example → texture*     **Ours**, *example → texture*

*Query Image*     *GT 3D model*     ***Top7 Retrieval Results***

Figure 3: **Qualitative Results.** Top: Texture synthesis from random codes or example images. Top Right: Texture synthesis from example images. Bottom: Shape retrieval. BicycleGAN [41] is a SOTA method for supervised multimodal I2I translation. Unseen Pool: ShapeNet [35] Pool. Zoom in for better view.

latent code $z$ and $c$ to each intermediate layer of $G$ following [62]. For the attentive multi-view metric learning (AMV-ML) part, we take the convolutional part of ResNet-34 as the backbone.

**Training Details.** We first train TSM, and utilize the learned $G$ to generate hard triplets in an online manner. We use the Adam solver [27] with a learning rate of 0.0002 and coefficients of (0.5, 0.999). The latent dimension $|z|$ is set to 8, and $|c|$ is the number of shape categories. We train the network with a image size of $256 \times 256$ and with a batch size of 16. For the AMV-ML procedure, we train the network in two stages, *i.e.,* a warm-up stage with $\mathcal{L}_{inst}$ for 20 epochs and a fine-tuning stage with the objective $\mathcal{L}_{AMV-ML}$ for another 60 epochs. We use the SGD solver with a beginning learning rate of 0.002, the momentum of 0.9, and the weight decay of 0.0004. The learning rate is fixed in the initial 10 \ 20 epochs, and linearly decays to zero over the next 10 \ 40 epochs. The model is trained with a batch size of 24 and with a image size of $224 \times 224$.

**Evaluation.** For evaluation, we render multi-view images without texture $\hat{S}$ for each 3D model contained in the 3D pool. The distance between a query image $X^i$ and a 3D shape $j$ can be computed as $\sum_{i=1}^{V} w_v^i * (\mathcal{F}(X^i) \odot \mathcal{F}(\hat{s}_v^j))$, where $\hat{s}_v^j$ is the $v$th view image of $\hat{S}^j$. See Sec. 2.2 (VG) for details. We use TopK Recall as our primary quantitative measure, while also report HAU (mean modified Hausdorff Distance) and IoU (Intersection Over Union) as suggested in [15].

### 3.2 Benchmark Performance

**Pix3D.** Pix3D [2] offers 10,069 images with 395 3D models. Following [15], we conduct experiments on categories that contain more than 300 non-occluded and non-truncated samples, *i.e.,* bed, chair, sofa, and table. This results in 5,118 images and 322 shapes, with 2,648 for training and 2,470 for test. We introduce 18,648 3D models from ShapeNet [35] as the 3D pool for more challenging evaluation.

**Stanford and Comp Cars.** Stanford Cars [3] focuses on more challenging fine-grained retrieval, where 134 3D car models with 16,185 images are provided. The images are officially split into the train (8,144) and test (8,041) sets. Comp Cars [4] shows similar characteristics as Stanford Cars. There are 94 3D car models with 5,696 images (3,798 for training and 1,898 for test). The 7,497 3D car models in ShapeNet [35] are used to demonstrate the robustness of our method.

**3D-FUTURE.** 3D-FUTURE [1] is a recently released 3D furniture repository. We study the IBSR subject on the six furniture categories, *i.e.,* sofa, bed, table, chair, cabinet, and stool. We have 25,913

Table 1: **Benchmark Performance.** For all the benchmarks, we train a single model for all categories (not category-specifc training). For Pix3D, We report both the category-specific scores and the mean scores as [15]. Here, "mean" represents the average scores of each category. For 3D-FUTURE, we conduct the sketch (Fig. 1) based method as our baseline. "mean*" shows the general performance over all the shapes (without considering the categories). Unseen Pool represents the ShapeNet [35] Pool.

| Method | Category | Seen Pool | | | | Unseen Pool | |
|---|---|---|---|---|---|---|---|
| | | Top1@R | Top10@R | HAU | IoU | HAU | IoU |
| **Pix3D** | | | | | | | |
| UDF-CGI [31] | bed | 19.4% | 46.6% | 0.0821 | 0.3397 | 0.0960 | 0.2487 |
| Grabner *et al.* [27] | | 35.1% | 83.2% | 0.0385 | 0.5598 | 0.0577 | 0.3013 |
| LFD [15] | | 64.4% | 89.0% | 0.0152 | 0.8074 | 0.0448 | 0.3490 |
| Ours | | **65.3%** | **95.4%** | **0.0122** | **0.8213** | **0.0425** | **0.3684** |
| UDF-CGI [31] | chair | 17.3% | 49.1% | 0.0559 | 0.3027 | 0.0843 | 0.1334 |
| Grabner *et al.* [27] | | 41.3% | 73.9% | 0.0305 | 0.5469 | 0.0502 | 0.1965 |
| LFD [15] | | 58.1% | 81.8% | 0.0170 | 0.7169 | 0.0375 | 0.2843 |
| Ours | | **87.9%** | **97.9%** | **0.0041** | **0.9063** | **0.0152** | **0.7482** |
| UDF-CGI [31] | sofa | 21.7% | 52.2% | 0.0503 | 0.3824 | 0.0590 | 0.3493 |
| Grabner *et al.* [27] | | 44.1% | 89.8% | 0.0197 | 0.7762 | 0.0294 | 0.6178 |
| LFD [15] | | 67.0% | 94.4% | 0.0075 | 0.9028 | 0.0178 | 0.7472 |
| Ours | | **72.8%** | **97.7%** | **0.0047** | **0.9070** | **0.0156** | **0.7963** |
| UDF-CGI [31] | table | 12.0% | 34.2% | 0.1003 | 0.1715 | 0.1239 | 0.1047 |
| Grabner *et al.* [27] | | 33.9% | 66.1% | 0.0607 | 0.4500 | 0.0753 | 0.1730 |
| LFD [15] | | 53.3% | 80.1% | 0.0288 | 0.6383 | 0.0482 | 0.2573 |
| Ours | | **73.7%** | **92.4%** | **0.0170** | **0.7667** | **0.0228** | **0.4391** |
| UDF-CGI [31] | mean | 17.6% | 45.5% | 0.0722 | 0.2991 | 0.0908 | 0.2090 |
| Grabner *et al.* [27] | | 38.6% | 78.3% | 0.0374 | 0.5832 | 0.0531 | 0.3222 |
| LFD [15] | | 60.7% | 86.3% | 0.0171 | 0.7663 | 0.0370 | 0.4095 |
| Ours | | **74.9%** | **95.8%** | **0.0095** | **0.8503** | **0.0240** | **0.6081** |
| **Stanford Cars** | | | | | | | |
| UDF-CGI [31] | car | 3.7% | 20.1% | 0.0198 | 0.7169 | 0.0242 | 0.6526 |
| Grabner *et al.* [27] | | 11.3% | 42.2% | 0.0153 | 0.7721 | 0.0183 | 0.7201 |
| LFD [15] | | 29.5% | 69.4% | 0.0110 | 0.8352 | 0.0150 | 0.7744 |
| Ours | | **68.4%** | **92.1%** | **0.0034** | **0.9210** | **0.0074** | **0.8735** |
| **Comp Cars** | | | | | | | |
| UDF-CGI [31] | car | 2.4% | 18.2% | 0.0207 | 0.7224 | 0.0271 | 0.6344 |
| Grabner *et al.* [27] | | 10.2% | 36.9% | 0.0158 | 0.7805 | 0.0194 | 0.7230 |
| LFD [15] | | 20.5% | 58.0% | 0.0133 | 0.8142 | 0.0165 | 0.7707 |
| Ours | | **67.1%** | **93.7%** | **0.0035** | **0.9256** | **0.0092** | **0.8591** |
| **3D-FUTURE (Mixed Pool)** | | | | | | | |
| Baseline | mean* | 31.8% | 65.4% | 0.05160 | 0.4915 | - | - |
| Ours | | **69.1%** | **91.7%** | **0.0198** | **0.7890** | - | - |

images and 4,662 3D shapes for our train set, and 5,865 images and 886 3D shapes for the validation set. In contrast to other datasets, the CAD models in 3D-FUTURE's validation set are unseen in the training stage. In the evaluation stage, we perform retrieval for the 5,865 images from the whole 5,548 (4,662 + 886) model pool. Comparisons on 3D-FUTURE are convincing due to its large 3D pool and the challenging setting.

**Performance.** The scores are reported in Table 1. Generally, our method acquires significantly higher accuracy compared against baselines and SOTA methods over all the metrics. For Pix3D, it (74.9%) outperforms the SOTA (60.7%) by a large margin. We notice that our approach captures similar scores as LFD [15] on the bed and sofa categories. The reason is that there are only 19 3D beds and 20 3D sofas in the pool. For Car benchmarks, our method can accurately identify the best-matched cars with an impressive Top1@R of 67.1% ∼ 68.4%, while previous SOTA scores are less than 30.0%. For the challenging scenarios on 3D-FUTURE, our method still yields a promising Top1 retrieval accuracy of 69.1%. Moreover, we evaluate our trained models on a large-scale unseen pool (ShapeNet). Our approach captures preferable IoU and HAU compared with previous works,

Table 2: **Hard Example Generation by Texture Synthesis.** The two baselines (Adaptation and 2.5D-Sketches) are illustrated in Fig. 1. For a fair comparison, all the experiments here are developed based on our AMV-ML in Sec. 2.2. Benefiting from AMV-ML, all the compared methods here performs much better than SOTA methods.

| Method | Top1@R | Top10@R | HAU | IoU |
|---|---|---|---|---|
| Pix3D (322 3D CAD models) | | | | |
| Adaptation | 68.5% | 94.5% | 0.0118 | 0.8050 |
| 2.5D-Sketches | 69.8% | 96.4% | 0.0104 | 0.8225 |
| Ours | **74.9%** | **95.8%** | **0.0095** | **0.8503** |
| Comp and Stanford Cars (210 3D Cars) | | | | |
| Adaptation | 56.5% | 87.1% | 0.0035 | 0.9020 |
| 2.5D-Sketches | 36.9% | 69.3% | 0.0078 | 0.8161 |
| Ours | **68.3%** | **93.2%** | **0.0028** | **0.9273** |
| 3D-FUTURE (5,548 3D CAD models) | | | | |
| Adaptation | 63.1% | 90.4% | 0.0242 | 0.7471 |
| 2.5D-Sketches | 31.8% | 65.4% | 0.0516 | 0.4915 |
| Ours | **69.1%** | **91.7%** | **0.0198** | **0.7890** |

Table 3: **Performance Gains of AMV-ML.** $\sqrt{}$: Optimizing the network via the selected loss. Gain: Top1@R gains compared with the baseline. We take $\mathcal{L}_{inst}$ as the baseline. $\mathcal{L}_{VG}$ and $\mathcal{L}_{view}$ are coupled, because $\mathcal{L}_{VG}$ relies on $\mathcal{L}_{view}$. See Sec. 2.2 for details of these loss items.

| $\mathcal{L}_{inst}$ | $\mathcal{L}_{SA}$ | $\mathcal{L}_{view} + \mathcal{L}_{VG}$ | Top1@R | HAU | IoU | Gain |
|---|---|---|---|---|---|---|
| $\sqrt{}$ | | | 56.3% | 0.0270 | 0.7077 | - |
| $\sqrt{}$ | $\sqrt{}$ | | 63.4% | 0.0234 | 0.7456 | +7.1% |
| $\sqrt{}$ | | $\sqrt{}$ | 61.4% | 0.0238 | 0.7354 | +5.1% |
| $\sqrt{}$ | $\sqrt{}$ | $\sqrt{}$ | **69.1%** | **0.0198** | **0.7890** | +12.8% |

which shows the generalization (or robustness) of our models. We can conclude from the surprising performance that hard sample generation by texture synthesis may be a potential venue for IBSR.

## 3.3 Ablation Studies

**Hard Example Generation.** We are the first to study hard triplets generation by texture synthesis for IBSR specific to its particular properties. We take Adaptation and 2.5D-Sketches in Fig. 1 as our compared methods. They can represent the two widely studied frameworks for IBSR. Adaptation here can be seen as a weakened version of our approach, where the positive and negative samples are textureless renderings. For 2.5D-Sketches, we firstly predict 2.5D sketches (*i.e.*, surface normal and depth) for anchor images via UperNet-50 [63]. Then, we render multi-view 2.5D sketches for each 3D CAD model. Finally, the positive and negative examples are 2.5D sketches. We combine Stanford Cars and Comp Cars to obtain an enlarged fine-grain car dataset with 210 (134+94-22) 3D cars. There are 22 identical 3D cars.

From Table 2, our method ranks 1st over all the metrics. In detail, our approach improves Adaptation by promising margins on all the scenarios (6.0%∼11.8%). The observation greatly supports our motivation that exploiting hard example generation by texture synthesis would improve the shape similarity learning for IBSR. Besides, 2.5D-Sketches may suffer from inaccurate dense predictions. It shows a performance gap compared with Adaptation and our methods in fine-grained retrieval. Finally, we find that $L_{cls}$ can help generate more visually appealing translations, but only improve the Top1 retrieval accuracy by 2.0% on 3D-FUTURE. One of the possible reasons is that when considering hard triplets or suppressing the adverse impact of texture like our method, the color information may be more important than the semantics of the texture.

**Attentive Multi-view Metric Learning (AMV-ML).** We discuss several baselines in Table 3 and Table 4 to study our AMV-ML on 3D-FUTURE. From the scores in Table 3, we can generally conclude that the proposed attention mechanisms are potent in remedying the background and viewpoint variation issues. From the top part of Table 4, the simple saliency attention mechanism can effectively improve the baseline by 7.7%. Coming to the bottom part, the proposed Viewpoint Guidance (VG) (69.1%) is a better option compared with Mean-Pooling (66.1%) and Max-Pooling

Table 4: **Attention Mechanisms in AMV-ML.** Our hard triplets generation policy is used in all the experiments. †: We take a mean pooling operation on the multi-view embedding of a 3D shape as its shape embedding (only in the evaluation phase). Mean-Pooling \ Max-Pooling: The embedding for a 3D shape is obtained by the mean \ max pooling operation in both training and evaluation stages.

| Method | Top1@R | HAU | IoU | Gain |
|---|---|---|---|---|
| Saliency Attention ($\mathcal{L}_{SA}$) | | | | |
| AMV-ML w/o $L_{SA}$ | 61.4% | 0.0238 | 0.7354 | - |
| AMV-ML | **69.1%** | **0.0198** | **0.7980** | +7.7% |
| Viewpoint Guidance ($\mathcal{L}_{view} + \mathcal{L}_{VG}$) | | | | |
| AMV-ML† w/o VG | 63.4% | 0.0234 | 0.7456 | - |
| Mean-Pooling | 66.1% | 0.0218 | 0.7685 | +2.7% |
| Max-Pooling | 64.7% | 0.0224 | 0.7572 | +1.3% |
| AMV-ML | **69.1%** | **0.0198** | **0.7980** | **+5.7%** |
| AMV-ML† | 67.2% | 0.0214 | 0.7758 | +3.7% |

(64.7%) in mitigating the influence of unconstrained viewpoints. Note that, [2] have demonstrated that directly adding pose supervisions to their network would largely degrade their retrieval performance. Here, we partially rectify the issue via our viewpoint guidance policy. Furthermore, we find that AMV-ML† (a degraded version) has a promising Top1@R of 67.2%. This observation shows that our framework is robust in enforcing the embedding network to capture the highlight shape characteristics.

Moreover, we discuss the impact of viewpoint distributions for the viewpoint guidance mechanism. For the indoor benchmarks (Pix3D and 3D-FUTURE), most of images (70.3%) are under the front viewpoints. The elevations mainly range from 20 to 45 degrees. The viewpoint distribution makes sense since most people prefer to captured indoor images from front views. For the outdoor Car benchmarks [3,4], the azimuth distribution follows a uniform distribution. We evaluate our trained models on the "Front" (5 views) and "Back" (5 views) viewpoints, while remove the east and west views. We find that our models are not sensitive to the azimuth distribution, though the azimuths of the training images are biased towards "Front" (Pix3D: 79.5% vs. 75.3%, 3D-FUTURE: 69.4% vs. 69.4%). Besides, some extremely unpreferable evaluates may decrease the retrieval performance, but our models show robustness in most cases.

## 4 Conclusion

In the paper, we studied metric learning for cross-domain image-based shape retrieval (IBSR). We found that the shape difference between a negative pair is entangled with the texture gap, making metric learning ineffective in pushing away negative pairs. We thus proposed to create hard triplets by texture synthesis, and developed a geometry-focused multi-view metric learning framework to tackle the issue. Our method effectively suppresses the impact of texture, thereby push the network to focus more on discovering geometric characteristics. Retrieval performances on different challenging benchmarks, including 3D-FUTURE, Pix3D, Stanford Cars, and Comp Cars, demonstrate the superiority of our approach. We also conduct various ablations to discuss our method more comprehensively. Our solution may be a potent venue for high-performing IBSR studies.

## Broader Impact Statement

Based on our knowledge, our work may not have an adverse impact on ethical aspects and future societal consequences. With the growing number of 3D shapes, the studies of cross-domain image-based shape retrieval (IBSR) is significant. We thus believe our work in this paper may have a positive impact on related subjects and techniques. For example, it can help to build 3D virtual scenes for real-world houses by accurately identify the exact 3D shapes contained in the captured 2D scene images. Furthermore, designers may be able to develop their required 3D CAD models based on the retrieved highly similar 3D shapes instead of drawing 3D shapes from scratches. High-performing IBSR systems may also inspire and benefit studies in 3D object reconstruction from large-scale shape collections.

## Acknowledgments

Dacheng Tao is supported by Australian Research Council Project FL-170100117.

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
