[Supplementary Material]

# Supplemental Materials for "Hard Example Generation by Texture Synthesis for Cross-domain Shape Similarity Learning"

**Huan Fu**[1*]    **Shunming Li**[1*]    **Rongfei Jia**[1]    **Mingming Gong**[2]

**Binqiang Zhao**[1]    **Dacheng Tao**[3]

[1]TaoXi Technology Department, Alibaba Group, China
[2]The University of Melbourne, Australia
[3]The University of Sydney, Australia

{fuhuan.fh, shunming.lsm, rongfei.jrf, binqiang.zhao}@alibaba-inc.com
mingming.gong@unimelb.edu.au    dacheng.tao@sydney.edu.au

## 1   Network Architecture

Figure 1: (a). Generator in Texture Synthesis Module (TSM). We show how to inject $z$ and $c$ into the generator, where $z$ and $c$ represents a latent code and a semantic latent code, respectively. Note that, for Stanford Cars and Comp Cars, $c$ has been ignored since there is only one category in the two benchmarks. (b). ResBlock of the texture encoder in TSM. (c) The Unet downsampling block. (d) The Unet upsampling block. More details are reported in Table 1.

The network architectures for our geometry-focused multi-view metric learning framework are reported in Table 1 (TSM), Table 2 (AMV-ML), and Figure 1. For convenience, we use the following abbreviation: $C_{in}$ = Input channel, $C_{out}$ = Feature channel, K = Kernel size, S = Stride Size, Conv =

---

Convolutional layer, FC = Fully-connected layer, ResBlock = A residual block, ResNet34-Conv$i_x$ = the $i$th convolutional block of ResNet34.

Table 1: The network architecture of our **Texture Synthesis Module (TSM)**. Here, $C$ is the category number of the dataset.

| Texture Synthesis Module (TSM) | | | | | |
|---|---|---|---|---|---|
| **Encoder** | | | | | |
| Index | Layer | $C_{in}$ | $C_{out}$ | K | S |
| 1 | Conv + InstanceNorm + LeakyReLU | 3 | 64 | 4 | 2 |
| 2 | ResBlock + InstanceNorm + LeakyReLU | 64 | 64 | 3 | 1 |
| 3 | ResBlock + InstanceNorm + LeakyReLU | 64 | 128 | 3 | 1 |
| 4 | ResBlock + InstanceNorm + LeakyReLU | 128 | 192 | 3 | 1 |
| 5 | ResBlock + InstanceNorm + LeakyReLU | 192 | 256 | 3 | 1 |
| 6 | ResBlock + LeakyReLU | 256 | 256 | 3 | 1 |
| 7 | Average Pool | - | - | 8 | 8 |
| 8 | Embedding $\mathcal{N}$ Linear $256 \rightarrow 8$ | - | - | - | - |
| **Generator** | | | | | |
| 1 | Conv + LeakyReLU | 3+4+8 | 64 | 4 | 2 |
| 2 | UnetBlock Down | 64+4+8 | 128 | 4 | 2 |
| 3 | UnetBlock Down | 128+4+8 | 256 | 4 | 2 |
| 4 | UnetBlock Down | 256+4+8 | 512 | 4 | 2 |
| 5 | UnetBlock Down x4 | 512+4+8 | 512 | 4 | 2 |
| 6 | Conv + ReLU | 512 | 512 | 4 | 2 |
| 7 | UnetBlock Up x 4 | 1024 | 512 | 4 | 2 |
| 13 | UnetBlock Up | 1024 | 256 | 4 | 2 |
| 14 | UnetBlock Up | 512 | 128 | 4 | 2 |
| 15 | UnetBlock Up | 256 | 64 | 4 | 2 |
| 16 | UnetBlock Up + Tanh | 128 | 3 | 4 | 2 |
| **Discriminator - 70x70** | | | | | |
| 1 | Conv + LeakyReLU | 3 | 64 | 4 | 2 |
| 2 | Conv + InstanceNorm + LeakyReLU | 64 | 128 | 4 | 2 |
| 3 | Conv + InstanceNorm + LeakyReLU | 128 | 256 | 4 | 2 |
| 4 | Conv + InstanceNorm + LeakyReLU | 256 | 512 | 4 | 1 |
| 5 | Conv | 512 | 1 | 4 | 1 |
| 5 | Average Pool + FC | 512 | $C$ | - | - |
| **Discriminator - 140x140** | | | | | |
| 1 | Average Pool | - | - | 3 | 2 |
| 2 | Conv + LeakyReLU | 3 | 32 | 4 | 2 |
| 3 | Conv + InstanceNorm + LeakyReLU | 32 | 64 | 4 | 2 |
| 4 | Conv + InstanceNorm + LeakyReLU | 64 | 128 | 4 | 2 |
| 5 | Conv + InstanceNorm + LeakyReLU | 128 | 256 | 4 | 1 |
| 6 | Conv | 256 | 1 | 4 | 1 |
| 6 | Average Pool + FC | 512 | $C$ | - | - |

## 2 Dataset Statistics

In Table 3, we report the statistics of the used datasets in our submission. For all the datasets, we train a single model instead of performing category-specific training. Note that, 3D-FUTURE here contains more fine-grained 3D CAD models. In contrast to other datasets, the 866 3D CAD models corresponding to the 5865 test images are totally unseen during the training procedure. In the evaluation stage, we use the full evaluation set (5548) as the 3D pool. Note that, Pix3D only provides a few 3D beds and sofas. Thus, our Top1@R is only slightly higher than previous methods on the bed and sofa categories.

Table 2: The network architecture of our **Attentive Multi-View Metric Learning (AMV-ML)**. For the Upernet decoder, see their public codes for more details.

| Index | Layer | $C_{in}$ | $C_{out}$ | K | S |
|---|---|---|---|---|---|
| \multicolumn{6}{c}{Attentive Multi-View Metric Learning (AMV-ML)} |||||
| \multicolumn{6}{c}{$Enc1$} |||||
| 1 | Conv + BatchNorm + ReLU | 3 | 64 | 7 | 2 |
| 2 | MaxPool | - | - | 3 | 2 |
| 3 | ResNet34 - Conv2$_x$ | 64 | 64 | 3 | - |
| \multicolumn{6}{c}{$Enc2$ and $Enc3$} |||||
| 1 | ResNet34 - Conv3$_x$ | 64 | 128 | 3 | - |
| 2 | ResNet34 - Conv4$_x$ | 128 | 256 | 3 | - |
| 3 | ResNet34 - Conv5$_x$ | 256 | 512 | 3 | - |
| \multicolumn{6}{c}{UperNet} |||||
| 1 | Adaptive Average Pool | - | - | - | - |
| 2 | Conv + BatchNorm + ReLU | 512 | 128 | 1 | 1 |
| 3 | Conv + BatchNorm + ReLU | 512+128 | 128 | 1 | 1 |
| 4 | Feature Pyramid Network | - | - | - | - |
| 5 | Conv + BatchNorm + ReLU | 512 | 128 | 3 | 1 |
| 6 | Conv | 128 | 1 | 1 | 1 |

Table 3: **Dataset Statistics.**

| Dataset | Category | Train Images | Train Models | Test Images | Test Models | Evaluation 3D Pool |
|---|---|---|---|---|---|---|
| Pix3D | bed | 198 | 19 | 196 | - | 19 |
| | chair | 1507 | 221 | 1387 | - | 221 |
| | sofa | 558 | 20 | 534 | - | 20 |
| | table | 384 | 62 | 354 | - | 62 |
| | total | 2647 | 322 | 2471 | - | 322 |
| Stanford | car | 8144 | 134 | 8041 | - | 134 |
| Comp | car | 3798 | 98 | 1898 | - | 98 |
| 3D-FUTURE | total | 25913 | 4662 | 5865 | 886 | 5548 (4662 + 886) |

## 3 Qualitative Results

We make qualitative comparisons with two widely studied retrieval solutions, including 2.5D-Sketch and Feature Adaptation (no-texture), on 3D-FUTURE. The results are shown in Figure 2. We can see that our method potentially focuses more on discovering the shape characteristics, thus achieve high-performing fine-grained retrieval. Note that, the other two methods are building on our proposed AMV-ML for fair comparisons, thus can also obtain reasonable retrieval results. More results are also shown in Figure 3.

We also report some challenging cases and failure cases in Figure 4. Firstly, we show three representative challenging cases on 3D-FUTURE, including partially occlusions, slight object incompletion, and unfavorable illumination. Our method can still capture acceptable retrieval sequences in these cases. However, our method can not handle some cases well, especially when the interested objects are heavily occluded, or the saliency objects are visually indistinguishable.

Figure 2: Qualitative comparisons with 2.5D-Sketch and Adaptation. For a fair comparison, all the experiments here are developed based on our AMV-ML in Sec. 2.2 in our submission. Benefiting from AMV-ML, all the compared methods here performs much better than SOTA methods.

Figure 3: **Qualitative Results on Other Public Benchmarks.**

Figure 4: Some Challenging cases and Failure cases on 3D-FUTURE.