[Reviews · NeurIPS 2020]

Review 1

Summary and Contributions: The authors propose a framework for image based 3d model retrieval by generating hard examples that help a feature network ignore effects of texture whilst attending to the geometric information. They achieve this by synthesizing negative examples using texture information from anchor images.

Strengths: The manuscript presents a through experimental evaluation of the framework: 1. The effectiveness of the approach is demonstrated against various benchmark datasets highlighting its applicability to wide range of categories. Additionally, comparisons are also provided with number of prior art. Significant improvement in retrieval rate and IoU is demonstrated. 2. Detailed ablations studies are provided to motivate the need for each of the design choices. 3. The idea to generate negative samples that share similar texture features to the anchor but differ in geometric details is a simple and elegant idea encourage the embedding network to rely on more geometric information rather than building a space that relies on texture features. 4. The use of saliency attention to weight the importance of features from the 2D image is an interesting design choice to eliminate background information from influencing the embedding network. 5. The datasets used have been adequately explained and the authors have highlighted the interesting features of each of the dataset that would make it easier/ harder for a IBSR framework to retrieve models from the dataset. 6. The supplementary materials have a thorough treatment of each of the networks used in the framework. And the experiment section also provides all the required hyper-parameters aiding in the reproducibility of the results. 7. The supplementary also provides a number of qualitative examples to demonstrate the effectiveness of the approach on a wide variety of geometries. Particularly, challenging and failure cases are also highlighted which helps provide better insights about the limitations of the approach

Weaknesses: Although the authors present an interesting idea, the manuscript would benefit from addressing some of the below concerns: 1. Major language and syntactical corrections are needed in many sections. Some sections are laden with typographical errors such as follows: line 92 : differs T .. -> differs from T .. line 118 : we going a further step -> we go a step further line 130 : element wise production -> element wise product? line 135 : In specific, -> Specifically, .. line 142 : We are now have an query .. -> We now have a query .. line 198 : an query -> a query Although this does not take away from the technical novelty of the paper, appropriate corrections are necessary for the manuscript to be ready for publication 2. The saliency attention method would benefit from additional explanation. Particularly, it is not clear what the F_4x, F_8x, F_16x and F_32x are. Are these the feature maps from different layers of a pretrained network? 3. It is not clear how the view point collection is used in tandem with the corresponding image. Particularly, explanation of the multi-view image collection would be instructive. Are these fixed set of views for all geometries. If so what are the range of elevations on the rendered images since the AMV-ML network only has an azimuth classifier. 4. It appears that most of the improvements come from L_view and L_VG. Particularly, adding viewpoint discrimination ability to the embedding network seems to improve retrieval rate significantly. The authors mention this observation in the experiment section. However, the chosen viewpoints seems like a function of the datasets. How would this loss help in more "in-the-wild" images whose viewpoint distribution might not be similar to the training dateset viewpoint distribution. 5. Extending on the point above, what is the best heuristic for choosing the viewpoints for any given dataset?

Correctness: The experimental design is consistent with prior art and the evaluation protocol is also appropriate for the given task.

Clarity: The paper presents an interesting idea but could use significant proof reading. Particularly, certain syntactical errors in certain sections as pointed out above needs to be corrected. Notwithstanding the concerns about the language, the remainder of the paper is well written and explains the task and solution well.

Relation to Prior Work: Adequate context is provided in relation to prior work and all the distinguishing features of this work in relation to prior art is appropriately highlighted. The experimental evaluation section also describes difference to comparative approaches in detail.

Reproducibility: Yes

Additional Feedback: Please correct syntactical and typographical errors as pointed out above. %%% Post Rebuttal: The authors address most of the concerns raised by the reviewers and make a case for the motivation of the proposed solution. In light of the rebuttal in am inclined to retain my original score of acceptance.


Review 2

Summary and Contributions: The authors propose a 2D to 3D shape retrieval system, where a 2D image is used to retrieve its 3D shape, that uses synthetic generation to learn how to separate texture from shape features. The retrieval technique relies on encoding both natural query images and 3D shapes to an embedding space and performing multi-view metric learning to learn the embedding space. The system generates hard triplets, where the positive sample has the same shape but different texture as the reference sample and the negative sample has different shape but the same texture. The authors make use of a conditional generative adversarial network to learn this model using a generator that takes as input an texture code and a class code as conditional variables for generating a 2D textured image. They apply their approach to three benchmarks: Pix3D, Stanford Cars, and Comp Cars. The main contribution is that the authors consider texture information generate hard triplets based on the properties of the reference image.

Strengths: - the authors propose a method that learns to disentangle shape from texture through a conditional GAN approach that relies on synthesizing images of 3D shapes - the approach is well motivated and is described clearly - the authors perform ablation analysis to demonstrate that various sub-components of their proposed approach contribute to the overall performance (saliency attention, viewpoint guidance) - the authors compare to various competing algorithms and show improvements on multiple datasets

Weaknesses: - some parts of the paper are not entirely clear. For example, what does "Specific to the particular properties of the IBSR subject" mean? Does it mean that texture generation is based on the properties of the 3D model? Or does it mean that texture generation is based on properties of the object in a 2D image? I think the problem is that "subject" is not defined here

Correctness: Appears to be correct.

Clarity: Yes

Relation to Prior Work: Yes, the paper clearly describes its relationship to previous work.

Reproducibility: Yes

Additional Feedback: None. POST REBUTTAL: After considering the author rebuttal and the other reviews, my original assessment has not changed. On novelty, my take is that conceptually the overall idea of using conditional GANs to separate shape from texture has not been done, so it is a novel model for the task, even if the general techniques used are not themselves novel (e.g., VAE's, GANs). The authors do make some modifications to conditional VAE-GANs that help in their scenario, but I didn't consider that to be a major contribution in my original review. I feel like the authors make a reasonable argument for why their submission is appropriate for NeurIPS.


Review 3

Summary and Contributions: This paper introduces a framework to improve upon previous image-based shape retrieval methods. This framework improves retrieval results by: 1. Using a generative network to synthesize hard examples, where the negative shape is rendered with similar texture as the query image. 2. Use mask (saliency) and viewpoint predictions to prune out superficial information. The authors compared the results of their proposed framework to several baselines and achieve superior results. Also, the author performed ablation studies to justify the usage of a generative network and viewpoint prediction in the proposed framework.

Strengths: This paper brings various experiments and ablation studies to confirm the effectiveness of their method. Specifically, the authors evaluated their method on four major datasets and compared with recent baselines.

Weaknesses: A major concern of mine is that this paper seems to be focused on a very specific problem (image-based shape retrival), that is loosely related to the NIPS community. The authors made significant efforts to bring SOTA results on this tasks, yet it is unclear how this could be of broader interests among the NIPS audience, especially the methods used in this paper is not very novel. I would suggest 3D Vision or SGP as better venues for this paper, where this topic is more widely appreciated. In terms of contribution and novelty, this paper does improve upon previous works and brings good performance, but the novelty of its solution is rather limited. The frame work is composed of several previous works with special adaptation to this specific problem.

Correctness: The empirical methodology and claims are sound. The major claim of this paper is that by rendering hard triplets, the feature extracted would be more related to geometric properties. This claim is supported by their evaluation on various datasets and through a ablation study.

Clarity: I think the paper is not very well written, with some awkard sentences, redundant words, typos, overloaded notations, and an over-complicated naming strategy. Here's a few examples: Overloaded notation: The normal distribution shares the same symbol as normal maps; The probability notion P is not in italic but the positive sample P is in italic. Awkward sentences: L34, 'Hypothesizing' has the wrong subject("Negative example"). Typos: L164, linear instead of liner. redudant words: L19. The object in the query image, the word "contained" is unnecessary. Naming Strategy: I think it is unnecessary to use the word saliency, where masks or silhouettes are better choices. Saliency is commonly used as regions important to human perception, which is not the case in this paper. Also I would not use the word 'attention' for the viewpoint module, as it is simply a weighted sum. Others: Eqn (9) should be a matrix-vector product instead. in table 2, 2.5D sketches has better top10@R in pix3D, but is not in bold.

Relation to Prior Work: The author makes considerable efforts to diffentiate their work from previous ones, which should be well received by the reader.

Reproducibility: Yes

Additional Feedback: ****** post rebutall ****** I do apologize for weighing the relevance issue too much. My major concern is that this paper would only meet the novelty bar if this problem is very relevant, as the major contributions are tightly bounded with the IBSR task. I also defend my comments on the use terms. 'Saliency' in the R.fig1 is a bad example for justifying the word saliency. It is ambiguous in a way that it could be the sofa in the back or the chair in the front. The word is tightly related to how human perceive objects and is usually a distribution. Therefore, by saying the saliency mask of R.Fig.1, it is ambiguous as it could be the sofa or the chair in the front. The word silhouette does not suffer from this problem, as it is bound to a specific object, i.e. the silhouette of the chair, the silhouette of the sofa. But the saliency mask of the image is definitely not a binary mask of a specific object: you will get pixels lighting up accross multiple objects. By referring to attention, I think the comments in my original feedback is not accurate. My concern is that it is masking in nature, but not the usual attention mechanism people refer to, which is not a good reading experience. In addition, it seems the authors do not commit to change other aspects of the writing in the rebutal; I assume this is simply due to the issue of sapce. I would strongly suggest the author to rework the writing, either for the CR version or for the next submission. I would also suggest the author not using pharses such as 'our clever methods' in the rebuttal. It is just not professional and might hurt your results in the end. All that being said, I would like to revise my score to slightly above the threshold, based on the assumption that this paper is very relevant, and the intrinsic work it has done, regardless of its flaws in writing & presentation, which I assume the author would fix if the paper got accepted.


Review 4

Summary and Contributions: In this paper, the author introduces a new approach for the cross-domain image-based 3D shape retrieval. Main contributions 1)Using the hard triplets generated by texture as the data to suppress the adverse effects of texture in 2D images, thereby push the network to focus more on discovering geometric characteristics, 2) Uses a clear mask attention mechanism to enhance the feature saliency regions, 3) Proposed a simple but effective viewpoint attention policy to guide the multi-view embedding aggregation.

Strengths: The paper is well written and present with a nice flow to explain clearly its contributions for cross-domain shape similarity learning (i.e. image, sketch, 3D shape domains). It is interesting to see that the paper uses a GAN-based idea for texture information extraction and then generate a hard triplet for similarity measurement. The paper including figures are well-prepared that help me understand the difference between the proposed approach and prior works (even though it is not a complete summary)

Weaknesses: The paper introduces some practical applications based on popular adversarial tricks for cross-domain (2D to 3D) instance corresponding (i.e. retrieval, reconstruction). Though presented with a nice flow with the demonstration of some interesting experimental results, it lacks the significant intellectual merits to make this paper under the bar of NeurIPS venue. The paper should be well fit WACV as an application paper or 3DV where a more relevant audience will appreciate the efforts for cross-domain analysis. Some comments on the paper: The author only compares the performance with the image retrieval, but there are many studies about the sketch retrieval (e.g. Deep Correlated Holistic Metric Learning for Sketch-based 3D Shape Retrieval, ). I recommend the author to test the performance of sketch retrieval using the same way as the paper introduced. The input dataset is very similar to the data of 3D reconstruction, the author should explain the difference between them.

Correctness: Yes

Clarity: Yes

Relation to Prior Work: Yes the figure 1 helps a lot to highlight the difference

Reproducibility: Yes

Additional Feedback: Post-rebuttal: I appreciate the authors' feedback and other reviewers' comments about this work. I agree that It is a good trick (generation of hard negative samples) to make the triplet network to focus on the shape of the input images other than the texture of the image during the similarity learning process. However, in general this method itself is an application of GAN to remove the texture of the shape. This is a fundamental application of GAN and widely used in many previous works, i.e. domain adaption/image-to-image translation. As suggested by other fellow reviewers ["[1]Texture Fields: Learning Texture Representations in Function Space, ICCV 2019"], The similar idea is adding the Saliency Attention. We can also use GAN to mark the background and get the shape information. It is a fancy application but using gan for domain translation is not novel. In addition, on the triplet loss is also a very widely used structure for this task. I will slightly increase my score.

[Author Response · NeurIPS 2020]



**R.Figure 1**

| ViewPoint | Front | Back |
|---|---|---|
| Pix3D | 79.5% | 75.3% |
| 3D-FUTURE | 69.4% | 69.4% |

*Image*  *Recon.*

*R#3:* Silhouette or Saliency? (object occlusions)   *R#4:* SBSR vs. IBSR   *R#4:* 3D Reconstruction Data

*R#1: Viewpoints.* We evaluate our trained models on the Front (5 views) and Back (5 views) viewpoints. We remove the east and west views. We find our trained models are not sensitive to the azimuths, though 70% of images are captured under front viewpoints.

*Online images.* Unpreferable elevations will decrease the performance. Our methods show robustness on most cases.

– ***To R#3 and R#4***. The reviewers **might misunderstand** our motivation and primary contribution when evaluating the paper. The comments such as "Taking efforts to bring SOTA results (R#3)" and "introducing practical applications based on popular adversarial tricks for IBSR (#4)" are the inaccurate understandings of our motivation and idea.

For IBSR, the main challenge is the large appearance gap between 3D shapes and 2D images. The common routine is to map 2D images and 3D shapes into an embedding space and learn a shape similarity measure. We **find that** the shape difference between a negative pair is entangled with the texture gap, making previous metric learning based methods ineffective in pushing away negative pairs. To tackle the issue, we **propose** an elegant idea (as stated by R#1 and R#2) to create hard triplets via the exploration of texture synthesis based on the properties of the IBSR subject. The generation of a hard triplet is shown in Fig. 1, **where** the positive example and the anchor are identical in shape (geometric details), but differ in texture; the negative example and the anchor are similar in texture, but differ in shape. We generate hard triplets in an online manner to improve the cross-domain shape similarity learning. Our **second contribution** is to introduce the saliency attention and viewpoint guidance mechanisms to remedy clutter background noises and unconstrained viewpoints issues of 2D nature images. **Both our motivation and idea are novel** and have been intensively studied in the experimental section. We believe our clever idea, *i.e.,* **generating texture to suppress the adverse impacts of texture**, is potent for future IBSR studies. Besides, our method ranks **1st** on the 3D-FUTURE AI Challenge (IBSR Track) (28 submissions on the leaderboard).

– ***Connection to the NeurIPS community (R#3 & R#4)***. **Firstly**, NeurIPS accepts subjects such as Computer Vision, Application, Information Retrieval, and Embedding Approaches. **Secondly**, IBSR is a fundamental subject in 3D Vision, and the community has put many efforts in building 3D datasets to support the studies of IBSR [1,2,3,4,59,60]. **Thirdly**, with the growing number of 3D shapes, the studies of IBSR is significant. For example, it can help to build 3D virtual scenes for real-world houses by accurately identify the exact 3D shapes contained in the captured 2D scene images. High-performing IBSR systems may also inspire and benefit studies of 3D object reconstruction from large-scale shape collections. Regarding its significance, the retrieval accuracy of IBSR is not that promising than the counterpart, *i.e.*, image retrieval. We thus believe our work in this paper benefits to the advancement of the research on the subject.

– ***Misunderstandings (R#3 & R#4)***. "The major claim of this paper is that by **rendering hard triplets** (**R#3**)": We **have not** claimed to "render" hard triplets. In fact, we can not render hard triplets since each 3D model contains a single UV atlas (texture) in the datasets. 90% of models in ShapeNet [35] are without texture. "**Silhouettes is better than Saliency** (**R#3**)": See R.Fig. 1, we care about the sofa instead of the coffee table. Since both the objects have their own "Silhouettes", "Saliency" is a better choice. There are lots of occluded objects [1,2]. "I would not use the word '**attention**' for the viewpoint module, as it is **simply a weighted sum.** (**R#3**)": Firstly, we use "guidance" instead of "attention". Secondly, most of "attention" strategies can be concluded as weighted sum operations. "**The paper only compares performance with IBSR methods.** (**R#4**)": Firstly, We study IBSR instead of sketch-based shape retrieval (SBSR). Secondly, IBSR focus more on the fine-grained geometric differences, while SBSR retrieves roughly similar shapes in category level. "**Explain the differences between the input datasets and 3D reconstruction datasets.** (**R#4**)": These datasets can be used for 3D reconstruction studies. However, researchers prefer to use rendered synthetic images (without backgrounds and occlusions) based on ShapeNet [35] for the 3D reconstruction subject.

– ***Syntactical corrections and suggested additional explanation for the saliency attention module (SAM) (R#1)***. Thanks for the constructive suggestions. We will correct these syntactical errors. For the SAM, the feature maps are the side features of the ResNet blocks ($\text{Conv}_{4x} \sim \text{Conv}_{32x}$) as presented in Supp-Tab. 2. We will make it clear in the paper.

– ***More discussions for the Viewpoint Guidance (VG) mechanism (R#1)***. For the indoor benchmark (Pix3D [2] and 3D-FUTURE [1]), we find that 70.3% of images are under the front viewpoints. The elevations mainly range from 20 to 45 degrees. The viewpoint distribution makes sense since most people prefer to captured indoor images from front views. For the outdoor Car benchmarks [3,4], the azimuth distribution follows a uniform distribution. We only study azimuths to make consistency with multi-view 3D representations. Our final version will address all the suggestions.

– ***Suggest to make the "particular properties" clear (R#2)***. Thanks for the constructive suggestion. The particular properties are: (1) In contrast to category-driven metric learning for image retrieval, each 3D shape is an individual instance (category). It's not easy to apply hard sample mining strategies in image retrieval to IBSR. (2) 2D objects with different appearances (in practice) may correspond to one 3D shape. (3) While appearance information is a strong feature for 2D image understanding, it has adverse impacts for IBSR. IBSR cares about the geometric details. Based on the properties and our observations, we believe our idea may be a potent venue for future high-performing IBSR studies.

[Meta-Review · NeurIPS 2020]

This is a borderline case since it applies the principles of "ImageNet-trained CNNs are biased towards texture; increasing shape bias improves accuracy and robustness" to 3D shape retrieval using conditional GANs in a way that is mostly straightforward. However, I would vote for the acceptance of this paper, given that the majority of the reviewers support the paper since "it was not done before", " the presented approach would serve as a strong baseline against most future approaches addressing the IBSR task", and "thoroughness of the experiments".